

# Disentangling the rates of carbonyl sulphide (COS) production and consumption and their dependency with soil properties across biomes and land use types

Aurore Kaisermann[1], Jérôme Ogée[1], Joana Sauze[1], Steve Wohl[1], Sam P. Jones[1], Ana Gutierrez[1], Lisa Wingate[1]

[1] INRA/Bordeaux Science Agro, UMR 1391 ISPA, Villenave d'Ornon, 33140, France

*Correspondence to*: Aurore Kaisermann (aurore.kaisermann@inra.fr)

**Abstract.** Soils both emit and consume the trace gas carbonyl sulphide (COS) leading to a soil-air COS exchange rate that is
the net result of two opposing fluxes. Partitioning these two gross fluxes and understanding their drivers are necessary to estimate the contribution of soils to the current and future atmospheric budget of COS.

Previous efforts to disentangle the gross COS fluxes from soils have used flux measurements on air-dried soils as a proxy for the COS emission rates of moist soils. However, this method implicitly assumes that COS uptake becomes negligible and COS emission remains steady while soils are drying. We tested this assumption by estimating simultaneously the soil COS sources
and sinks and their temperature sensitivity ($Q_{10}$) from soil-air COS flux measurements on fresh soils at different COS concentrations and two soil temperatures. Measurements were performed on 27 European soils from different biomes and land use types in order to obtain a large range of physical-chemical properties and identify the drivers of COS consumption and production rates.

We found that COS production rates from moist and air-dried soils were not significantly different for a given soil and that the
COS production rates had $Q_{10}$ values (3.96 ± 3.94) that were larger and more variable than the $Q_{10}$ for COS consumption (1.17 ± 0.27). COS production generally contributed less to the net flux that was dominated by gross COS consumption but this contribution of COS production increased rapidly at higher temperature, lower soil moisture and lower COS concentrations. Consequently, measurements at higher COS concentrations (*viz.* 1000 ppt) always increased the robustness of COS consumption estimates. Across the range of biomes and land use types, COS production rates co-varied with total soil
nitrogen ($r = 0.68$, $P < 0.05$) and the first-order COS uptake rate co-varied most with microbial N content ($r = 0.64$, $P < 0.05$) providing new insights on how to upscale the contribution of soils to the global COS atmospheric budget.



## 1 Introduction

Carbonyl sulphide (COS) is a powerful greenhouse gas whose atmospheric concentration has varied considerably during the Earth's history (Ueno et al., 2009). Nowadays, the concentration of COS in the atmosphere is around 500 ppt (or pmol mol$^{-1}$) almost 1 million times less than current atmospheric $CO_2$ concentrations. However, the relative seasonal amplitude of COS is

about 5 times larger than that of $CO_2$ and has led to COS being proposed as a powerful tracer of gross primary production (GPP) over land {e.g. Montzka et al., 2007; Sandoval-Soto et al., 2005; Whelan et al., 2017}. This is because the removal of COS from the atmosphere during summer months in the Northern hemisphere is driven by the activity of plants over land that remove COS through an enzymatic reaction with carbonic anhydrase (CA) (Sandoval-Soto et al., 2005; Stimler et al., 2012), an ubiquitous enzyme particularly abundant in leaf mesophyll cells (Fabre et al., 2007).

Soils are also teeming with a diverse range of organisms such as bacteria, fungi and algae that also contain CAs (Elleuche and Pöggeler, 2010; Moroney et al., 2001; Smith and Ferry, 2000) and thus have the potential to remove COS from the atmosphere. For this reason, oxic soils are generally considered net sinks for atmospheric COS (Van Diest and Kesselmeier, 2008; Kesselmeier et al., 1999; Li et al., 2005; Whelan et al., 2016) albeit with a much weaker sink strength than vegetation (Berry et al., 2013; Campbell et al., 2017; Kettle et al., 2002; Launois et al., 2015). Some soils have also been found to be strong

producers of COS, notably anoxic soils (Fried et al., 1993; Hines and Morrison, 1992; Whelan et al., 2013) in addition to some oxic soils with and without plant litter (Bunk et al., 2017; Kitz et al., 2017; Maseyk et al., 2014; Melillo et al., 1993; Sun et al., 2016; Whelan et al., 2016; Whelan and Rhew, 2015).

Despite the importance of soil COS fluxes for improving the global COS mass budget, the mechanisms underlying COS production and consumption by soils remain unclear. This is partly because it is difficult to disentangle the functional response

of these two opposing fluxes in isolation and characterise how changes in the environment or soil properties impact the net COS flux. Recently physical and enzymatic models describing the consumption of COS by soils have been advanced (Ogée et al., 2016; Sun et al., 2015). However, equivalent mechanistic understanding of their COS production is still lacking.

One approach for estimating COS emission rates from soils is to measure the net COS flux rate of air-dried soil samples (Whelan and Rhew, 2016; Whelan et al., 2016). This assumes that the COS consumption by dry soils is negligible as hydrolysis

by CA requires the presence of water to proceed (Van Diest and Kesselmeier, 2008; Ogée et al., 2016; Sun et al., 2015; Whelan and Rhew, 2016). Thus with a further assumption that COS emission rates do not vary while soils are drying, COS emission rates can be retrieved from the net COS flux measured on fresh soils (Whelan et al., 2016). However, it is still not clear whether the COS production by soils is related to biological activity and potentially varying with soil moisture. If this was the case, this method for estimating COS production rates could create strong biases in the partitioning of the net COS flux under wet

or moist field conditions.

An alternative approach, presented by Conrad (1994), facilitates the concomitant estimation of COS production and consumption on fresh soils, thereby providing a test of whether COS production changes with soil water content or not. Because this alternative approach requires the measurement of net COS fluxes at different atmospheric COS concentrations, it cannot





be easily implemented in the field without large artefacts (Castro and Galloway, 1991; Mello and Hines, 1994), but it is well adapted to measurements on soil microcosms (Lehmann and Conrad, 1996). So far very few studies have implemented this approach, however those that have, always used very high COS concentrations (> 100 times greater than current atmospheric levels), thus the partitioning of COS fluxes at ambient concentrations still remains unexplored (Lehmann and Conrad, 1996).

In this study, we used the approach of Conrad (1994) to concomitantly estimate COS production and consumption rates simultaneously from moist soils near ambient COS concentrations. We combined this approach with a new theoretical framework (Ogée et al., 2016) to retrieve COS production and first-order consumption rates on a range of soils from different biomes and land use types located in Europe and Israel. We then evaluated the effects of varying COS concentration, soil moisture and temperature on the COS production and consumption rates and assessed the potential role of soil properties such

as pH, texture, soil carbon and nitrogen, microbial biomass, etc., as drivers of COS gross fluxes.

## 2 Materials and methods

### 2.1 Theory

Assuming that the soil-air COS exchange rate in soil microcosms is governed by only three processes, namely diffusion through the soil column, production and uptake *via* hydrolysis, Ogée *et al.* (2016) proposed a steady-state, analytical model of the COS

efflux at the soil surface ($F$, pmol m$^{-2}$ s$^{-1}$) as follows:

$$F = \left( -\sqrt{k\,B\,\theta\,D}\;C_a + \sqrt{\frac{D}{k\,B\,\theta}}\,\rho_b\,P \right) \tanh\left( \frac{1}{z_{max}} \sqrt{\frac{k\,B\,\theta}{D}} \right), \qquad (1)$$

where $D$ (m$^2$ s$^{-1}$) is the COS diffusivity through the soil matrix, $k$ (s$^{-1}$) is the first-order COS hydrolysis rate constant and $P$ (pmol kg$^{-1}$ s$^{-1}$) is the COS production rate. Other symbols are $B$ (m$^3$ m$^{-3}$), the COS solubility in soil water, $\theta$ (m$^3$ m$^{-3}$), the volumetric water content, $C_a$ (mol m$^{-3}$), the *molar* concentration of COS in the air at the soil surface, $\rho_b$ (kg m$^{-3}$), the soil bulk

density and $z_{max}$ (m), the maximum soil depth. This equation assumes a finite soil depth ($z_{max}$) and uniform soil properties ($\theta$, $\rho_b$ …) and is therefore only suited for soil microcosms studies (Ogée et al., 2016).

When soil moisture tends to zero ($\theta \rightarrow 0$), Eq. 1 simplifies to $F_{dry} = \rho_b P_{dry}/z_{max}$. where $F_{dry}$ and $P_{dry}$ represent the net COS flux $F$ and the COS production rate $P$ of a air-dry soil, respectively. Thus, assuming that $P$ does not vary with soil moisture ($P = P_{dry}$), the COS production rate can be estimated from measurements of the soil-to-air COS flux performed on air-dry soil

samples. This method, hereafter called Method 1, has been proposed recently for partitioning soil COS sources and sinks (Whelan et al., 2016). By knowing the value of $P$, the COS hydrolysis rate constant $k$ can then be estimated using Eq. 1, from measurements of the soil-to-air COS flux performed on moist soils. In practice a numerical iterative method must be implemented to find the value of $k$ that minimises the discrepancy between the observed and modelled fluxes (Ogée et al., 2016; Sauze et al., 2017a).



Note that if the steady-state soil COS flux $F$ is measured at different COS concentrations $C_a$, and provided that $P$ is known, it is possible to derive a different $k$ value for each ($F$, $C_a$) pair of measurements. In theory each pair of measurements should provide the same hydrolysis rate constant, unless COS hydrolysis does not follow first-order kinetics and $k$ is not a true rate constant and varies with $C_a$. In fact this might well be the case as $k$ was originally hypothesised to be a catalysed rate constant

(Kesselmeier et al., 1999; Lehmann and Conrad, 1996) that follows Michaelis-Menten kinetics (Ogée et al., 2016; Protoschill-Krebs and Kesselmeier, 1992). In this case the hydrolysis rate would saturate to a maximal rate $V_{max}$ at very high COS concentrations and would respond linearly to $C_a$ with a slope $k = V_{max}/K_m$ only at low concentrations, i.e., when $C_a << K_m$ where $K_m$ (mol m$^{-3}$) is the so-called Michaelis-Menten coefficient and corresponds to the COS concentration at which $k$ equals $k_{max}/2$.

Carbonic anhydrases (CA) are a family of enzymes ubiquitous in soil micro-organisms that are known to catalyse COS hydrolysis (Protoschill-Krebs et al., 1992; 1996). The exact values of their kinetic parameters are very scarce but tend to show relatively high values of $K_m$, around 40-60 µM at 20°C (Haritos and Dojchinov, 2005; Ogawa et al, 2013; Ogee et al., 2016; Protoschill-Krebs and Kesselmeier, 1992), i.e., more than 2 million times the atmospheric COS concentration (500 ppt or 20 pM). With such high $K_m$ values, and assuming that COS consumption by soils is only caused by CA-driven hydrolysis, the

rate constant $k$ should be well approximated by $k_{max}/K_m$ and thus $F$ should respond linearly to $C_a$ (see Eq. 1), without any sign of saturation. A near-linear response of $F$ to $C_a$ has indeed been demonstrated on all soils tested so far, even at COS concentrations 100 times higher than ambient levels (Lehmann and Conrad, 1996). Based on this observation, Eq. 1 can be re-written:

$$F = F_0 - V_{d0}C_a, \tag{2}$$

where $F_0$ is the gross flux of COS production, i.e., the flux $F$ when $C_a = 0$, and $V_{d0}$ (m s$^{-1}$) is the COS deposition velocity onto the soil surface that would occur in the absence of COS production (i.e. if $P = 0$). The gross flux of COS uptake is calculated by subtracting $F_0$ from the net COS flux $F$.

From Eq. 2 we can see that, by performing steady-state COS flux measurements at different COS concentrations, it is possible to estimate simultaneously $V_{d0}$ and $F_0$ from the slope and the intercept of a linear regression between the two variables, leading

to the determination of $P$ and $k$:

$$F_0 = \sqrt{\frac{D}{k\,B\,\theta}}\,\rho_b\,P\,\tanh\left(\frac{1}{z_{max}}\sqrt{\frac{k\,B\,\theta}{D}}\right), \tag{3a}$$

$$V_{d0} = \sqrt{k\,B\,\theta\,D}\,\tanh\left(\frac{1}{z_{max}}\sqrt{\frac{k\,B\,\theta}{D}}\right), \tag{3b}$$

In practice, this method, hereafter called Method 2, is performed in three steps. First a 3-point linear regression is performed

between $F$ and $C_a$ data to estimate $F_0$ and $V_{d0}$ for each microcosm (reference for the linear regression R package). Then the value of $k$ that satisfies Eq. 3b is obtained using an iterative numerical method (reference for the root finding R package).





Finally this $k$ value is introduced into Eq. 3a to estimate $P$ from $F_0$. Values for $B$ are estimated from soil temperature using Wilhelm et al. (1977) and the COS effective diffusivity $D$ is estimated using the empirical formulation of Moldrup et al. (2003) for repacked soils (see also Ogée et al., 2016).

## 2.2 Soil sampling and preparation

Soils from 27 locations were collected along a latitudinal gradient in Europe and Israel during the summer of 2016. These locations were selected to cover a range of biomes and land use as well as soil physico-chemical properties (see Supplement Table S1). The first 10 centimetres of the soil were collected in sealed plastic bags and sent to INRA Bordeaux after collection. Upon reception, the different soils were sieved using a 4mm mesh, homogenised and stored at 4°C separated in two batches: one batch was air-dried for 1-2 weeks before being used to estimate the air-dried COS production rate ($P_{dry}$) hereafter referred

to as "dry" and another batch was stored at 4°C for several weeks until it was used to estimate the COS production and consumption rates of fresh soils ($P$ and $k$) hereafter referred to as "moist".

For both methods, measurements were performed in triplicates, using soil microcosms consisting of custom-made glass jars (0.825 dm$^3$ volume, 8.85 cm internal diameter). For Method 1, 350-400 g of air-dried soil was used in each microcosm and acclimated in the dark at 18° C in a climate-controlled chamber (MD1400, Snijders, Tillburg, NL) for 2-3 days prior to the gas

exchange measurements. For Method 2, 200-300 g of equivalent dry soil were used and maintained at $17.7°C \pm 0.4°C$ and 30% of maximum water holding capacity (WHC) for 2 weeks until the gas exchange measurements were performed. Prior to this incubation period, the WHC of each soil was estimated using the method of Haney and Haney (2010) and, when fresh soils were too wet (> 30% WHC), they were air-dried until they reached 10-20% WHC, then re-humidified to 30% WHC at the start of the 2-week incubation period. Throughout this incubation period, moisture contents were monitored gravimetrically

and maintained by adding distilled water.

## 2.3 Gas-exchange measurements

Glass soil microcosms were equipped with screw-tight glass lids equipped with two stainless steel fittings Swagelok® (Swagelok, Solon, OH, USA) to connect to the 1/8" Teflon inlet and outlet lines of the measurement system. A stainless steel temperature probe (3-wire PT100, 15 cm length, 3 mm diameter, reference RS 362 9935) continuously recorded the average

soil temperature in each microcosm. Dry synthetic air was adjusted to the desired $CO_2$ ($399 \pm 6$ ppm) and COS mixing ratios and supplied to the microcosms using the same system as described in Gimeno et al. (2017). The inlet and outlet airstreams of each microcosm were analysed sequentially using a mid-infrared quantum cascade laser spectrometer (QCLS, Aerodyne Research Inc Billerica, MA, USA), coupled upstream to a Nafion dryer (MD-070-24-S-2, Perma Pure LLC, Lakewood, NJ, USA) to remove matrix effects caused by water vapour (Kooijmans et al., 2016). To account for instrument drift, an auto-

background was implemented regularly (typically every 38 minutes) for 120 s using a dry $N_2$ bottle. A 2-point calibration scheme was also implemented using the same dry $N_2$ bottle (measured every 14 minutes) and an Aculife-treated cylinder (Air Liquide USA, Houston, TX, USA) filled with compressed air and 524.8 pmol(COS) mol$^{-1}$ calibrated to the NOAA-Scripps



Institution of Oceanography provisional scale. This second cylinder was measured every 14 to 56 minutes depending on the sequence used.

Using a custom-made multiplexed system (Sauze et al., 2017a), six jars with six different soils and one empty jar (blank) were measured sequentially over *ca.* 18h to investigate simultaneously 6 different soils under identical conditions (Supplement Fig. S2). Over this period the measuring sequence consisted of 8 steps that measured the COS fluxes from all the microcosms at 2 different temperatures (18°C and 23°C) and 3 different COS concentration levels (around 100, 500 and 1000 ppt), with an acclimation time of *ca.* 2 hours following a change in temperature and 40 minutes following a change in COS concentration (see Supplement Fig. S2). While only 14 mins were usually required to stabilise the COS mixing ratio on the chamber lines after a step change in the COS mixing ratio of the inlet line, two hours seemed the minimum time required to stabilize the soil temperature to a new temperature.

For each temperature and COS concentration level, three inlet/outlet pairs were measured on each microcosm. Each line was measured for 120 s and only the last 15 s were retained to compute the mean COS concentration, accounting for the residence time of air in the tubing and gas analyser. The median standard deviation during these last 15s was 12.4 ppt for COS and 0.09 ppm of $CO_2$. From each inlet/outlet pair the soil-to-air COS flux was computed as follows:

$$F = \frac{\phi}{S}(c_a - c_{in}) \ , \tag{4}$$

where $F$ is the COS flux (pmol m$^{-2}$ s$^{-1}$), $\phi$ is the flow rate of dry air through the chamber (mol s$^{-1}$), $S$ (0.00615 m$^2$) is the soil surface area, $c_{in}$ (pmol mol$^{-1}$) is the COS mixing ratio on the inlet and $c_a$ (pmol mol$^{-1}$) is the COS mixing ratio on the outlet. The air flow rate $\phi$ was set at 0.250 nlpm, i.e., 186 µmol s$^{-1}$. The COS flux for the blank chamber was never significantly different from zero.

The molar COS concentration ($C_a$) was estimated from molar ratio ($c_a$) and soil temperature measurements using the ideal gas law and an air pressure of 106000 Pa. The slight over pressure in the glass jars (of about 5 kPa) had been estimated previously during a preliminary experiment using a pressure transducer (BME280; Bosch GmbH, Gerlingen, Germany).

**2.4 Estimation of soil COS production and hydrolysis rates**

The COS production rate was first estimated on air-dried soils at 18°C and under atmospheric concentration (*ca.* 500 ppt) levels of COS. This "dry" production rate ($P_{dry}$, pmol kg$^{-1}$ s$^{-1}$) was deduced from the COS flux (Eq. 4) according to:

$$P_{dry} = SF_{dry}/M_{dry} \tag{5}$$

where $M_{dry}$ (kg) is the mass of dry soil in the microcosm. The COS production and hydrolysis rates on fresh soils ($P_{moist}$ and $k_{moist}$, respectively) were estimated using COS flux measurements performed at the three COS concentrations and Eqs. 2 and 3 described above. The linear relationship between $F$ and $C_a$ observed over a wide range of COS concentrations was confirmed using our set-up over the range of COS mixing ratios used in our experiments, i.e., 0-1200 ppt (see Supplement Fig. S3). These results justified the use of only three COS levels (referred to as "low", "med" and "high" hereafter) to perform the linear regression and calculate $P_{moist}$ and $k_{moist}$ in subsequent analyses. The COS mixing ratio in the inlet airstream of each microcosm





was thus set to $1111 \pm 29$ ppt ("high"), $557 \pm 10$ ppt ("med") or $124 \pm 8$ ppt ("low"), while the $CO_2$ mixing ratio was always maintained around $399 \pm 6$ ppm.

In order to evaluate whether the method used to estimate the COS production rate influenced the calculation of the COS hydrolysis rate of moist soils, we also used $P_{dry}$ to re-calculate the hydrolysis rate of moist soils as in previous studies. To do so, we inserted $P_{dry}$ into Eq. 1 and solved for the hydrolysis rate that satisfied the equation for a given level of COS concentration (referred to as $k_{recal,low}$, $k_{recal,med}$ and $k_{recal,high}$ hereafter).

The COS production and hydrolysis rates for the wet soils ($P_{moist}$ and $k_{moist}$) were measured at two temperatures (18°C and 23°C) to estimate their temperature sensitivity ($Q_{10}$) in this temperature range:

$$Q_{10(k)} = \left(\frac{k_{moist(23°C)}}{k_{moist(18°C)}}\right)^2, \tag{6a}$$

$$Q_{10(P)} = \left(\frac{P_{moist(23°C)}}{P_{moist(18°C)}}\right)^2, \tag{6b}$$

### 2.5 Soil physico-chemical properties

At the end of each gas exchange measurement, the soils were analysed for a range of physico-chemical properties. Soil texture and total C, N and $CaCO_3$ contents were measured using standard procedures at the INRA soil analyses platform (http://www.lille.inra.fr/las). Soil pH and redox potential were measured using a 1:5 soil-water ratio. Bulk density was estimated from the weight and volume of each soil microcosm. Soil water content was estimated gravimetrically as the weight difference between moist and oven-dried soil extracts. The concentration of phosphate ions was measured as in Van Veldhoven and Mannaerts, (1987). Microbial biomasses for carbon (C) and nitrogen (N) were estimated as the difference of dissolved C and N contents between fumigated (24h of chloroform fumigation) and non-fumigated soil extracts consisting of 10g of soil mixed with 40ml of 0.5 M of $K_2SO_4$ and shaken for 30min.

### 2.6 Statistical analyses

All data processing and graphs were made with R software (Version 3.3.3, R core Team, 2015) using the packages dplyr, lubridate, data.table and ggplot2 to examine the biome and land use effects on the gross COS production ($P_{moist}$) and first-order hydrolysis rate ($k_{moist}$) constants, to assess whether the differences between $k_{recal,med}$ and $k_{moist}$ depended on atmospheric COS concentration and to compare the temperature response of $P_{moist}$ and $k_{moist}$ using ANOVA and Tukey's HSD tests. To investigate the correlation between soil properties and all COS fluxes (gross COS production and uptake, hydrolysis rate at 18°C) spearman coefficient correlations were calculated and test for significance were performed with the corrplot package (Wei and Simko, 2017).



# 3 Results

All moist soils were net COS sinks at 18°C, ranging in magnitude from -7.66 to -0.78 pmol m$^{-2}$ s$^{-1}$ (Fig. 1), while the uncertainty on the blank was only of -0.11 ± 0.24 pmol m$^{-2}$ s$^{-1}$. This variability across different land use types and biomes was not explained by any of the measured environmental variables. Using the theoretical framework presented above we partitioned

the net COS fluxes measured on moist soils to assess COS production and uptake rates and to compare moist ($P_{moist}$) with dry ($P_{dry}$) soil COS production rates. As illustrated in Fig. 2a the COS production rates measured on moist soils ($P_{moist}$) were not significantly different from those measured on dry soils ($P_{dry}$). Overall the relationship between $P_{moist}$ and $P_{dry}$ was highly significant ($P < 0.001$) and followed a linear regression slope of 0.98 with an intercept of 0.02 pmol kg$^{-1}$ s$^{-1}$ (Fig. 2a). Dispersion of data around the linear regression ($r^2 = 0.59$) indicated that some soils were occasionally underestimated by one

method compared to the other.

Our study also indicated that the rates of COS production from moist soils measured at 18°C ($P_{moist}$) were significantly higher in temperate regions compared to those measured in boreal and Mediterranean regions (ANOVA $P = 0.0009$, Tukey's HSD tests: Temperate-Mediterranean $P=0.0009$, Temperate-Boreal $P=0.03$, Mediterranean-Boreal $P=0.4$; Fig. S4). The highest

COS production rates were measured on soils coming from temperate grassland sites (Fig. S4). Further analysis indicated that the eight temperate soils exhibiting the highest COS production rates also contained high C and N contents (Fig. 2a). The total C and N contents of the different soils were positively correlated with high microbial C and N biomass as well as redox potential, whilst negatively correlated with bulk density (Figs. 3 and 4 and Supplementary Table S1). No significant effect was detectable between $P_{moist}$ and latitude, longitude or land use cover. However, COS production rates were significantly and

positively correlated with soil N content ($r = 0.68$) and soil redox potential ($r = 0.53$) and negatively correlated with pH ($r = -0.43$) (Figs. 3 and 4).
The partitioned gross COS uptake rates ($V_{d0}C_a$) measured at 30% WHC were always much larger in absolute values (between -7.66 and -1.34 pmol m$^2$ s$^{-1}$) than the COS production rates $P_{moist}$ (less than 0.81 pmol m$^2$ s$^{-1}$) and thus dominated the net COS flux $F$ (Supplement Fig. S4). The first-order COS hydrolysis rate constant $k_{moist}$ was estimated to vary between 0.05

to 0.47 s$^{-1}$ and the relationship between the two estimates, $k_{moist}$ and $k_{recal,mid}$, were strongly ($r^2 = 0.96$) and linearly related, exhibiting a slope and intercept of 0.94 and 0.02 s$^{-1}$, respectively (Fig. 2b). Although the use of $P_{moist}$ or $P_{dry}$ had little influence on the retrieval of the first-order COS hydrolysis rate constants (Fig. 2b), the relative difference between the COS hydrolysis rates $k_{moist}$ and those re-calculated using $P_{dry}$ and Eq. 1 were significantly different when measured across different COS concentrations (Fig. 5; $P = 0.002$). The COS hydrolysis rate that satisfied the equation for low COS concentrations, $k_{recal,low}$

(estimated using $F$ and $C_a$ data from the low COS concentration measurements) was significantly ($P = 0.0011$) lower than at high COS concentrations, $k_{recal,high}$ (estimated using $F$ and $C_a$ data from the high COS concentration measurements), while $k_{recal,mid}$ (estimated using the medium COS concentration measurements) was intermediate and not significantly different from either $k_{recal,high}$ ($P = 0.52$) or $k_{recal,low}$ ($P = 0.056$). In addition, $k_{recal,low}$ values exhibited a larger spread of the deviation from $k_{moist}$



than both $k_{recal,mid}$ and $k_{recal,high}$ (Fig. 5). This occurs because the estimation of the COS hydrolysis rate using Eq. 1 becomes more sensitive to the value of $P$ prescribed when flux measurements are performed at low COS concentrations as there is a decrease in the fraction of COS uptake with respect to COS production. In contrast to the results found for the COS production rates, the first-order COS hydrolysis rate constants $k_{moist}$ were not related to land use or biome (Figs. 3 and 4). On the other hand, $k_{moist}$ values were positively and significantly correlated with microbial N ($r = 0.64$) and C ($r = 0.45$) biomass contents (Figs. 2, 3 and 4).

The temperature sensitivity ($Q_{10}$) of $P_{moist}$ had a mean and standard deviation of $4.36 \pm 4.45$. This was significantly higher ($P < 0.0001$) than the $Q_{10}$ of the hydrolysis rate that had a mean and standard deviation of $1.26 \pm 0.29$ (Fig. 6). The variability in $Q_{10}$ values across the 27 soils was also much larger for COS production rates than for COS hydrolysis rates. The temperature sensitivity of $P_{moist}$ did not correlate with any of the measured soil properties (Figs. 3 and 4). However, the $Q_{10}$ values of the COS hydrolysis rate constants were significantly and negatively correlated with soil total C content ($r = -0.49$) and positively correlated to bulk density ($r = 0.36$) (Figs. 3 and 4).

## 4 Discussion

### 4.1 Are COS production rates measured on dry soils a reasonable proxy for those occurring in moist soils?

Net COS fluxes measured from oxic soils commonly exhibit a unimodal response to water-filled pore space (WFPS) (Kesselmeier, Teusch & Kuhn, 1996; Van Diest & Kesselmeier, 2007; Whelan et al., 2006). Recently a theoretical framework was advanced describing how WFPS influences the diffusion of COS in the soil matrix and how this partially regulates the rate of COS hydrolysis by the enzyme CA in addition to temperature and COS concentration (Ogée et al., 2016). Following this theoretical framework we estimated that by maintaining moisture levels in our soils at ~30% water holding capacity we would be conducting our experiments very close to the optimum WFPS (between 15 to 37%) for gross COS uptake. However, it was not clear whether COS production should respond to variable soil water content, for our soils thus it was important to test this assumption as this would hinder the use of dry soil COS production rates as a robust proxy for COS production rates expected at optimum WFPS in moist conditions. Our experimental results support the use of dry soil COS production rates as a proxy to infer COS production rates from moist soils under optimal moisture conditions (Fig 2a). A recent study by Bunk et al. (2017) performed net COS flux measurements over a range of WHC between 3 and 90% on two different soils treated with either the fungicide nystatin or the antibiotic streptomycin that are assumed to suppress COS uptake by fungi and bacteria, respectively. They found that, on one of the soils (a tropical soil from Suriname), the COS production rate (estimated as the net COS flux measured after the nystatin treatment) was not responsive to soil moisture providing support for our experimental results. However, Bunk et al. (2017) also found that another soil, a temperate agricultural soil from Germany, exhibited a soil moisture response both before and after the nystatin treatment suggesting that the observed soil moisture response may be strongly driven by the COS production flux rate, which is in contradiction with the current theory presented in Eq. 1. Indeed when the hydrolysis rate constant tends to zero ($k \rightarrow 0$), Eq. 1 simplifies to $F = \rho_b P / z_{max}$ so that the net COS flux $F$ should



become independent of soil moisture, as long as $P$ does not respond to soil moisture. Thus to reconcile with theory the results from the nystatin-treated agricultural soil of Bunk et al. (2017), we would need to invoke a partial and/or non-uniform inhibition of $k$ by the nystatin application. A non-uniform reduction of soil moisture upon drying could also create a soil moisture response without the need to evoke a dependence of $P$ on soil water availability. Interestingly, if we extrapolate the results of Bunk et al. (2017) at 0% WHC, the (fully-dry) net COS flux would correspond reasonably well to the nystatin-inhibited flux measured at 30%WHC, and would further corroborate the results presented in the current study (Fig. 2a).

A further result of our study showed that when using Eq. 1 assuming $P_{dry}$ as a proxy for $P$ to estimate the COS hydrolysis rate constants $k$, the uncertainty on $k$ increased at lower atmospheric COS concentrations (Fig. 5). This is because, as atmospheric COS concentration decreases, the contribution of the sink term to the net COS flux becomes progressively smaller. More importantly this increased uncertainty was biased towards smaller $k$ (Fig. 5), with a median $k$ significantly lower (around 20%) at $ca$. 100 ppt compared to that obtained at $ca$. 800 ppt. Even at $ca$. 800 ppt, the re-calculated $k$ ($k_{recal,high}$) was still on average smaller than $k_{moist}$ (Fig. 3). In addition, it is worth noting that the bias between $k_{recal}$ and $C_a$ cannot be explained by Michaelis-Menten kinetics. Indeed, according to such an enzymatic model, $k$ should remain constant for $C_a << K_m$ and should decrease, not increase, at higher $C_a$ concentrations. Thus, overall, our results suggest that studies using $P_{dry}$ as a proxy for $P$ (Whelan et al., 2016) can provide a good proxy for moist conditions but may lead to a small and probably not significant underestimation of the COS uptake rate constant if measurements are made over the range of COS concentrations tested in our study.

**4.2 Relative importance of gross COS production to the net soil COS flux**

Across a range of biomes and land use types we found that the relative contribution of COS emission to the net soil COS flux was generally smaller than the COS uptake rate, but increased at higher temperatures and lower atmospheric COS concentrations. At COS concentrations close to those found in the atmosphere (~500 ppt), net COS fluxes were always negative in our soils when measured at 18°C, indicating that the compensation point (*i.e.* the COS concentration at which the net flux is zero) was always below the atmospheric COS concentration. Even at 100 ppt, only five of the soils had positive net COS fluxes indicating that the COS compensation point was lower than 100 ppt for the majority of the soils. This is consistent with previous studies on oxic soils (Kesselmeier et al., 1999; Liu et al., 2010) but contradictory to the results of Lehman and Conrad (1996) who found much higher compensation points. This apparent contradiction might be explained by the fact that Lehman and Conrad explored a much higher and wider range of COS concentrations (60-410 ppb) where different COS consumption processes might take place (e.g. physio-sorption). Further studies conducted in sub-tropical monsoon humid climates have also reported COS compensation points above 100 ppt (Geng and Mu, 2004; Yi and Wang, 2011), but still below the atmospheric concentration (i.e. around 300 ppt, respectively). These higher compensation points might be explained by the warm temperatures expected in this type of climate that should favour COS production over consumption (Fig. 6 and S5), shifting the compensation point to higher concentrations and causing some soils to become net sources of COS to the atmosphere upon warming. Our finding is supported in the literature with a number of studies reporting temperature sensitivities of emission-dominated net COS soil fluxes in the range of 1.7 to 3.3 (Maseyk et al., 2014; Saito et al., 2002; Saito et al., 2002; Whelan and





Rhew, 2015, 2016). Altogether, these results support the importance of taking into account the strong variability in COS production contributing to the net COS flux across different biomes varying in soil temperature when scaling for atmospheric budgets.

### 4.3 Drivers and mechanisms of COS production across European soils from different biomes and land use

Currently, COS emissions by oxic soils are considered to be abiotic in origin (Kitz et al., 2017; Whelan and Rhew, 2015). Dramatic COS production rates have been observed across US and Chinese arable soils (Billesbach et al., 2014; Liu et al., 2010; Maseyk et al., 2014; Whelan and Rhew, 2015). However, the exact mechanisms underlying COS production are still under debate (Whelan et al., 2016). A number of hypotheses including the thermal degradation of soil organic matter or desorption of COS from soil surfaces have been proposed and are partially supported by the persistence of COS emissions

after autoclaving (Kato et al., 2008; Whelan & Rhew, 2015; Whelan et al., 2016). Another abiotic process that could lead to COS production is the chemical reaction that occurs in flue gas from molecules present during combustion such as $CH_4 + SO_2 \Leftrightarrow COS + H_2O + H_2$ (Rhodes et al., 2000). Both sulphur dioxide ($SO_2$) and methane ($CH_4$) can be produced in soils, however $CH_4$ is generally produced in anaerobic zones of submerged soils and tends not to accumulate at the soil surface (Le Mer and Roger, 2001). It is not clear whether this reaction would be possible in aerobic, dry soils. The thermal decomposition of

$CH_3SCO$ radicals (Barnes et al., 1994) and the oxidation of thioformaldehyde and DMS (Barnes et al., 1996) could also lead to the production of COS. However, these two reactions are unlikely to explain our results on dark-incubated soils, as both reactions require the photolysis or photoproduction of certain compounds for this reaction to proceed.

There is growing evidence that biotic processes may also contribute to observed COS emission rates (Whelan et al., 2017). In particular a number of studies provide direct evidence for the production of COS during the hydrolysis of thiocyanates when

catalysed by thiocyanate hydrolase, an enzyme found in a range of bacteria (Katayama et al., 1992; Kim and Katayama, 2000; Ogawa et al., 2013) and fungi (Masaki et al., 2016). If COS production rates were even partially driven by such biotic processes, this contribution might be sensitive to soil water content and expected to decrease at very low soil water content as microbial activity tends to slow down and microbes enter either a stationary growth phase and/or a dormant state (Roszak and Colwell, 1987). However, we did not observe any significant reduction in COS production rates after air drying of the soils (Fig. 2a).

One potential explanation for this could be that some microorganisms can persist for prolonged periods of time in drought conditions, utilising energy reserves at a very slow rate (Raubuch et al., 2002) but nonetheless remain metabolically active (Manina and McKinney, 2013). For example, Zoppini and Marxsen (2010) demonstrated that some extracellular activities in river sediments were not reduced even after one year of drying. This can arise as air-dried soils can still contain some residual water in soil micropores that maintain enzymatic activity. The amount of liquid water required for maintaining such biological

activity, including thiocyanate hydrolase activity, could be extremely small and still result in a detectable amount of COS emitted. In addition, Maire et al. (2013) showed that endoenzymes released from dead organisms were stabilised in soils and could still lead to extracellular oxidative metabolism. This could also partly explain the continuation of COS production even at very low water content in our soils. In this context, even sterilised (autoclaved) soils might still produce COS as microbial



death can release nutrients and intracellular metabolites into the soil environment, including endo-enzymes capable of resisting the autoclaving process. Therefore, it is possible that the COS emission rates on autoclaved soils might relate to past biotic activity. In this respect, an interesting result from our study was that the magnitude of COS emitted from soils was positively correlated to total N concentration (Figs. 2a, 3 and 4) over a range of soil N concentrations between 0.38 and 10.2 g kg$^{-1}$ (Table

S1). Although this is the first study to demonstrate a significant relationship between soil N concentration and gross COS production rates, previous studies have observed shifts in the magnitude of net COS and $CS_2$ fluxes upon fertilisation with nitrate in both deciduous and evergreen coniferous forests (Melillo & Steudler, 1989). In addition a number of studies on agricultural soils in the US and China have observed large temperature-sensitive emissions of COS (Billesbach et al., 2014; Liu et al., 2010; Maseyk et al., 2014; Whelan & Rhew, 2015). Currently, the mechanisms for the observed link between sulphur

and nitrogen cycling in soils is still not understood. However it is known that S-containing amino acids such as methionine, cystine and cysteine are all potential precursors of COS and $CS_2$ (Bremner & Steele, 1978; Minami & Fukushi, 1981a; Minami & Fukushi, 1981b). Soils exposed to higher nutrient inputs may thus contain soil organic matter with relatively more N-containing precursors available for either biotic or abiotic degradation. Clearly further studies investigating the link between soil N inputs and gaseous S emissions are now required. In the meantime, our study brings another element of understanding

and clearly demonstrates that soil N content and temperature could be the main drivers of the COS production rates observed in plant-free soils and thus provide a future modelling framework to elaborate the consequences for atmospheric chemistry at larger scales.

**4.4 Drivers and mechanisms of COS uptake by soils**

Direct evidence for the role of carbonic anhydrase (CA) in the uptake of COS has been established in past lab experiments

with plant extracts (Protoschill-Krebs et al., 1996) and indirectly on soils treated with CA inhibitors (Kesselmeier et al., 1999). Based on the theoretical framework that exists for the catalysis of $CO_2$ uptake by CA in soils (Wingate et al. 2010; Sauze et al., 2017b), Ogée et al. (2016) developed an analogous framework to describe the uptake of COS by CA in soils (Eq. 1) and was able to reproduce the observed response of the net and gross COS uptake rate with water-filled pore space and its optimum. Our study showed that the response of the net COS uptake to soil water content is dominated by changes in the gross COS

uptake, not the COS production rate, in agreement with Eq. 1.

An important parameter in this modelling framework is the temperature sensitivity ($Q_{10}$) of the CA-catalysed COS hydrolysis rate $k$. In the present study a mean value of $1.23 \pm 0.29$ was estimated for the $Q_{10}$ of $k_{moist}$ over the entire range of 27 soils and exhibited much lower variability than the temperature sensitivity response of gross COS production (Fig. 6). Although the range of $Q_{10}$ for the hydrolysis rate was linearly and negatively related to soil C content (Fig. 4), this parameter appeared fairly

conservative amongst the different soils, and its mean value was also consistent with a range of published $Q_{10}$ values (1.22 to 1.9) for plant CA extracts (Burnell and Hatch, 1988; Boyd et al., 2015, Ogée et al., 2016), reinforcing the idea that the uptake of COS by the soils studied is driven by CA activity.



The large scale variability in the COS hydrolysis rate (at a given temperature and 30%WHC) was mostly related to variations in microbial C biomass (Fig. 4), and the majority of the smaller COS hydrolysis rates were indeed found in soils with the lowest microbial biomass. This result is consistent with the model of Ogée et al. (2016) that proposes soil CA activity to vary proportionally to the total volume of all the microbes, present in a soil provided that their CA requirements are similar. Our study, in addition to two further field studies (Saito et al., 2002; Yi et al., 2007), provide support for such a hypothesis, although differences in pH (Ogée et al. 2016; Sauze et al., 2017b) and microbial community structure (Sauze et al. 2017b) may complicate the relationship between the COS uptake rate constant and microbial biomass.

## 5 Conclusions

Uncertainties in the contribution of oxic soils to the atmospheric mass balance are large, with estimates for the global soil sink strength varying from between 70 and 510 GgS y$^{-1}$ (Berry et al., 2013; Campbell et al., 2017; Kettle et al., 2002; Launois et al., 2015; Montzka et al., 2007; Suntharalingam et al., 2008). Although developments in the mechanistic understanding and modelling of soil-atmosphere COS modelling have been made recently (Ogée et al., 2016; Sun et al., 2016) it still remains a challenge to extend the observations of a limited set of experimentally different datasets to robust descriptions of soil-atmosphere COS exchange in land surface models. This lack of coherently collected data across multiple biomes and land use types currently hinders advances in modelling the variability in atmospheric COS concentrations at the large scale. Our study goes some way towards addressing this gap by providing a comprehensive dataset of partitioned COS fluxes across Europe and Israel alongside the prominent soil characteristics that are commonly measured and mapped, providing potential transfer functions that can translate soil physical and chemical properties into globally gridded maps of COS production and uptake by soils. In particular our observations that COS hydrolysis rates are linked to variations in microbial C biomass whilst COS production rates are linked to the variability in total soil N with both gross COS fluxes exhibiting distinct temperature and moisture sensitivities provides a promising avenue for constraining the global COS sink strength of soils and their contribution to the atmospheric mass budget.

## Author contribution

AK, JO, LW, JS designed the experiments and AK, JS, SW, SJ and AG carried them out. AK prepared the manuscript with contributions from JO and LW.

## Competing interests

The authors declare that they have no conflict of interest.





## Acknowledgements

This project has received funding from the European Research Council (ERC) under the European Union's Seventh Framework Programme (FP7/2007-2013) (grant agreement No. 338264, awarded to L.W), the French Agence Nationale de la Recherche (ANR) (Grant Agreement No. ANR-13-BS06- 0005-01) and the Institut National de la Recherche Agronomique (INRA) departments EFPA and EA (PhD studentship for JS). We are also grateful to all the scientific teams across Europe and Israel that provided their time to collect soils from their local experimental sites for this study especially Jorge Curiel-Yuste, Alexandria Correia, Jean-Marc Ourcival, Jukka Pumpanen, Huizhong Zhang, Carmen Emmel, Nina Buchmann, Sabina Keller, Irene Lehner, Anders Lindroth, Andreas Ibrom, Jens Schaarup Sorensen, Dan Yakir, Fulin Yang, Michal Helias, Susanne Burri, Penelope Serrano Ortiz, Maria Rosario Moya Jimenez, Jose Luis Vicente, Holger Tulp, Per Marklund, John Marshall, Nils Henriksson, Raquel Lobo de Vale, Lukas Siebicke, Bernard Longdoz, Pascal Courtois, Katja Klumpp.

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



| Site ID | Country | Site | Biome | Landuse | Latitude | Longitude |
|---|---|---|---|---|---|---|
| CH-Cha | Switzerland (CH) | Chamau | Temperate | Grassland | 47.2 | 8.4 |
| CH-Dav | Switzerland (CH) | Davos | Temperate | Ever.Forest | 46.8 | 9.9 |
| CH-Fru | Switzerland (CH) | Fruebuel | Temperate | Grassland | 47.1 | 8.5 |
| CH-Lag | Switzerland (CH) | Lageren | Temperate | Dec.Forest | 47.1 | 8.5 |
| CH-Oe2 | Switzerland (CH) | Oensingen | Temperate | Cropland | 47.3 | 7.7 |
| DE-Hai | Germany (DE) | Hainich | Temperate | Dec.Forest | 51.1 | 10.5 |
| DE-Lei | Germany (DE) | Leinefelde | Temperate | Dec.Forest | 51.3 | 10.4 |
| DK-Sor | Denmark (DK) | Soro | Temperate | Dec.Forest | 55.5 | 11.6 |
| ES-Amo | Spain (ES) | Amoladeras | Mediterranean | Semi-arid Grassland | 36.8 | -2.3 |
| ES-Bal | Spain (ES) | Balsablanca | Mediterranean | Semi-arid Grassland | 36.9 | -2.0 |
| ES-Ube1 | Spain (ES) | Ubeda_Veg | Mediterranean | Orchard | 37.9 | -3.2 |
| ES-Ube2 | Spain (ES) | Ubeda_noVeg | Mediterranean | Orchard | 37.9 | -3.2 |
| FI-Hyy | Finland (FI) | Hyytiala | Boreal | Ever.Forest | 61.8 | 24.3 |
| FI-Var1 | Finland (FI) | Varrio1 | Boreal | Ever.Forest | 67.8 | 29.6 |
| FI-Var2 | Finland (FI) | Varrio2 | Boreal | Ever.Forest | 67.8 | 29.6 |
| FR-Hes | France (FR) | Hesse | Temperate | Dec.Forest | 48.7 | 7.1 |
| FR-Laq1 | France (FR) | Laquielle1_Int | Temperate | Grassland | 45.6 | 2.7 |
| FR-Laq2 | France (FR) | Laquielle2_Ext | Temperate | Grassland | 45.6 | 2.7 |
| IL-Reh | Israel (IL) | Rehovot | Mediterranean | Orchard | 31.9 | 34.8 |
| IL-Yat | Israel (IL) | Yatir | Mediterranean | Ever.Forest | 31.3 | 35.1 |
| PT-Cor | Portugal (PT) | Coruche | Mediterranean | Ever.Forest | 39.1 | -8.3 |
| PT-Mit-b9 | Portugal (PT) | Mitra | Mediterranean | Ever.Forest | 38.5 | -8.0 |
| SE-Hyl | Sweden (SE) | Hyltemossa | Boreal | Peatland | 56.1 | 13.4 |
| SE-Nor | Sweden (SE) | Norunda | Boreal | Ever.Forest | 60.1 | 17.5 |
| SE-Ros_Cont | Sweden (SE) | Rosinedal_Cont | Boreal | Ever.Forest | 64.2 | 19.7 |
| SE-Ros_Fert | Sweden (SE) | Rosinedal_Fert | Boreal | Ever.Forest | 64.2 | 19.7 |
| SE-Sva | Sweden (SE) | Svartberget | Boreal | Ever.Forest | 64.2 | 19.8 |

5  **Table 1:** Locations and names of sites sampled across Europe and Israel describing climatic and land use characteristics



**Figure 1:** Variability in the net COS flux measured at 18°C and an atmospheric COS concentration of 500 ppt on replicated (n=3) moist (30% WHC) soils sampled from across Europe and Israel (see Table 1). The letters A denote agricultural sites and N denote those sites fertilised with nitrogen.





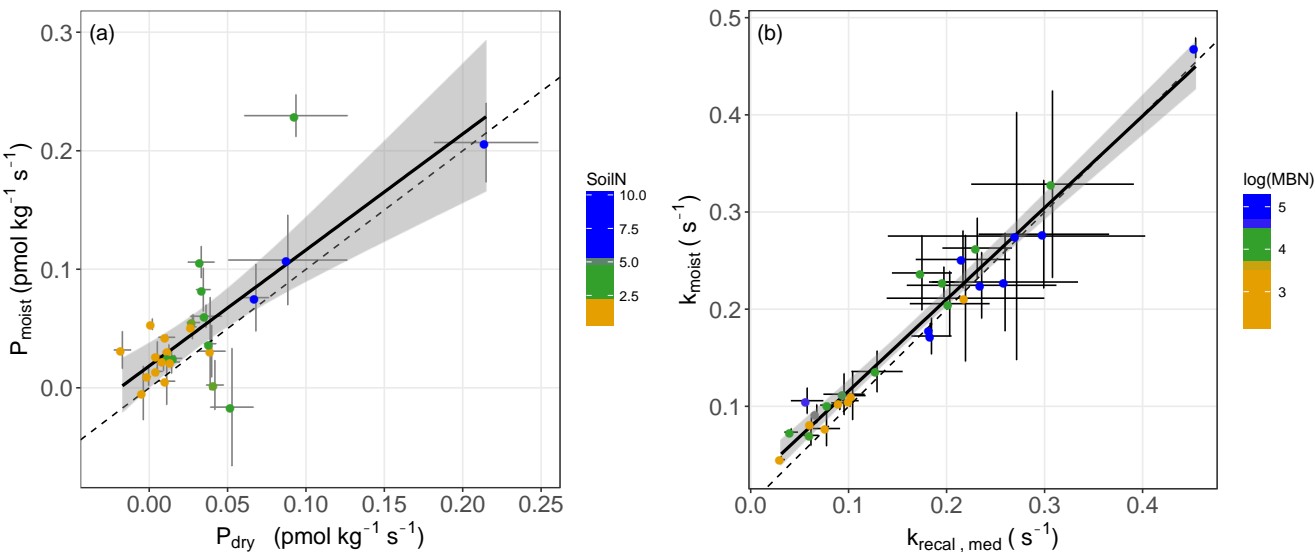

**Figure 2:** Comparison of (a) the soil COS production rates measured on air-dried soils ($P_{dry}$; method 1) and moist soils ($P_{moist}$; method 2) for the range of soil total nitrogen concentrations (g kg$^{-1}$) measured at each site and (b) the COS hydrolysis rate of moist soils ($k_{recal,med}$ and $k_{moist}$, respectively) for the range of microbial biomass nitrogen (µg g$^{-1}$) at each site. Each point represents the mean flux $\pm$ SD for each site measured at 18°C (n=3). The dashed lines represent the 1:1 slope and the solid black lines represent the slope of the linear models, the grey areas represent the 95% confidence level interval for predictions from the linear models.





**Figure 3:** Biplot principal component analysis (PCA) of the 27 soils in this study. Each small point represents the mean of the three replicates of one soil coloured by the biome (Boreal, Mediterranean, Temperate), the big points represent the barycentre of each biome. Black arrows are the active variables (standardized physico-chemical properties) used to build the PCA (BD= Bulk density; MBC and MBN = microbial biomass carbon and nitrogen; WFPS = water filled pore space).

5 To investigate the interrelations between COS fluxes and soil properties, variables of COS fluxes (source = gross COS source at 18°C; uptake = gross COS uptake at 18°C; Q10k and Q10P = $Q_{10}$ of hydrolysis rate and of the source; and k18= hydrolysis rate at 18°C) were fitted as supplementary variables into the PCA using the package R called FactoMineR. The purple arrows are the supplementary variables which the coordinates projected on the PCA are predicted using only the information provided by the performed PCA on active variables.

10 The principal component analysis of soil properties showed that the microbial biomass C is the parameters that contributed the most to the first principal component (15%), and that is positively correlated to soil C and N content, microbial biomass N and potential redox, while negatively correlated to bulk density (each contributing between 8 and 14% to the first principal component). The most contributing variables to the second principal component second axis were soil texture (sand, clay and silt).





**Figure 4:** Spearman correlation coefficients (rho) between soil properties (SoilN and SoilC are the soil N and C content, BD is bulk density, MBC and MBN are microbial biomass C and N, Pho is the phosphorus content) and COS fluxes (Source is the gross COS source at 18℃, Uptake is the gross COS uptake at 18℃, k18 is the hydrolysis rate at 18℃). Only significant correlations are shown (*P*<0.05).





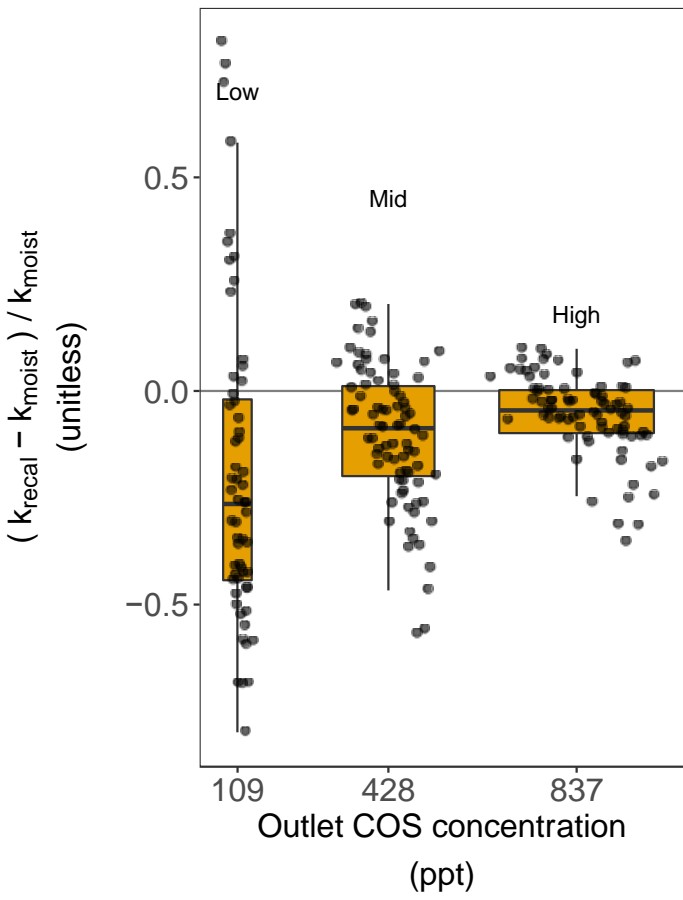

**Figure 5:** Difference in the COS hydrolysis rates of moist soils calculated using the production source estimated from dried soils ($k_{recal}$ with $P_{dry}$) and moist soils ($k_{moist}$ with $P_{moist}$) as a function of three different atmospheric COS concentrations in the outlet. Each grey point represents one soil replicate, with the red box showing the lower quartile, median and upper quartile values, the whiskers indicate the range of variation in the difference, and the box width represents the range of variation in the net COS flux measured for the different soils.





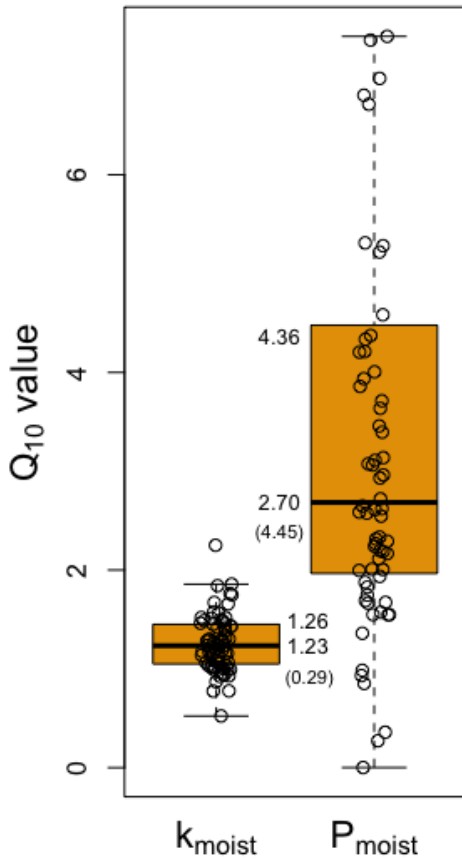

**Figure 6:** The estimated temperature sensitivity ($Q_{10}$) of COS production ($P_{moist}$) and hydrolysis rate ($k_{moist}$) across 27 sites in Europe and Israel. Each point represents the estimated parameter for each of the 3 replicated microcosms incubated at two temperatures from all sites. The box indicates the lower quartile, median and upper quartile values, the whiskers show the range of variation in the difference and displays the mean (SD) and median value for the $Q_{10}$ parameter.