# Peer review of "Disentangling the rates of carbonyl sulphide (COS) production and consumption and their dependency with soil properties across biomes and land use types"

_Atmospheric Chemistry and Physics, 2017_

## Referee Comment (RC1) · Anonymous Referee #1 · 3 Mar 2018

This is a nicely written paper with some important results for the carbonyl sulfide community.

Major comments: Pg 5, line 25: You describe using dry synthetic air flows for these experiments. Will the drying of the samples using these gas flows affect the results? Or was the air humidified and it's mentioned? Or was this accounted for by adding distilled water (as described in Pg 5 line20)? It's worth clarifying this.

Other general comments: I like the layout of the Discussion with the driving question

at the start but it might have even more impact with the answers to these questions instead. Might be nice to follow that through the Discussion rather than the current headers.

I feel like the abstract could do with a nice tie-up/bigger implications type sentence. Could you include something from the conclusions (maybe the N impact?)?

Minor comments:

Pg 1, line 25: Not essential. But could you edit the last sentence to be more specific? It seems really vague.

Pg 4, line 30: references missing.

Pg 5, line 8: Were the soil samples chilled for shipping? Or do you have any idea of the temperature history of the samples?

Pg 8, line 2: Do you mean the bias in the blank? The uncertainty on the blank value is 0.24 pmol.

Figure 1: I know it looks nice to have the fluxes in order but I'm trying to visualize what this is actually telling us. Would it be worthwhile grouping them by biosphere so they can be applied to other studies a little easier? So Boreal forest vs Peatland. Mediterranean Orchards vs grass vs forest and Temperate grass vs forest? Or something like that. And is grassland FR really boreal? Or do you mean alpine? Not in Table 1 so hard to tell. You could include the full site labels too. Could you also add some gridlines so it's easier to see what the labels are matching to?

Pg 10 line 25ish: Nice summary here.

Pg 11 line 4: I think this section could be tightened a little.

Figure 3: I must admit that I don't follow Fig 3. What do the dimensions (x vs y axis) represent? Is it really necessary?

Could you include Fig S4 in the main text? It's just missing the A and N labels for fertilizers. Is the CH Grassland not an agricultural land (Fig S5)? I's surprised you can find an unfertilized grassland in Europe!

Fig 4: Is there a reason for the order in Fig 4? Would it make more sense to keep the soil type (clay, silt, sand) together and microbial properties (Soil N, Soil C, Soil P, MBC and MBN) together? Then derived properties like PCOS (production), LCOS (loss) and k18. Some of the labels are not adequately explained. What do you mean by Redox, Q10k, etc. I know they are explained in the text but make sure the figures can be read independently.

Fig 5 and 6 could go in the supplement.

Table 1: Could you add the altitude of the sites? And maybe the annual mean soil temperature and moisture at each site if you have that data? Can you also include an explicit Fertilizer or not column?

---

## Referee Comment (RC2) · M.E. Whelan (Referee) · 27 Mar 2018

This is a well-conceived set of experiments to further our understanding of oxic soil OCS exchange. The approach to calculate the hydrolysis constant is commendable. It's also nice to see the 1996 Lehmann and Conrad study getting more use. Please note the follow up study in 2000 by Conrad and Meuser, "Soils contain more than one activity consuming carbonyl sulfide" Atmosphehric Environment, 21, 3635-3639.

Major Comments

[Figure]

P3:L22-25 I'm not sure how theta going to zero results in the simplified equation presented here. Are there some assumptions that need to be spelled out?

P5:L5-20 Please add more detail. For either method, were the jars partially sealed, generating higher $CO_2$ levels than ambient? Were any sensitivity studies performed – for example, did you find that incubating the soils for less than 2-3 days or two weeks led to different results? For method 2, were these soils kept in the dark as well? When soils were air dried, were they put into a jar or spread out in a pan for a more even drying? Sieving is an important choice here, too. Litter plays a role in surface OCS fluxes, sometimes contributing nearly all of the OCS uptake. Sieving removes most of the litter and soil structure. While we can't have everything in our experiment vary, it would be worth justifying the method approach a bit more. Regarding maintaining soil moisture by adding water – if the soil has dried out enough (probably crossing some threshold that has yet to be described) and water is added, the soil can experience a dramatic increase in OCS uptake that takes several hours to days to recover from. This is akin to the "Birch effect" for OCS. Please mention how much water was typically added to the soils. Did you see any decay curve in the soils that were maintained this way? Also, 18 hours is a long time to have dry air run over soils without substantial water loss. Were soils checked and re-watered during the incubations?

P7:L10 Is 5 degrees sufficient to calculate a meaningful Q10? Also, OCS uptake rates tend to exhibit a temperature optimum. The Q10 idea links the rate of reaction with a constant increase in rate with increase in temperature. Please justify the use of Q10. It would be good to know the natural variation in temperature of the sites as well.

P7:L15 I was not aware that soil redox potential could still yield a valid measurement after 2 weeks. How do you think this variable changed during the incubations themselves?

P8:L10 What is going on with the green points in Fig 2 that have a wide spread? Also, it appears that sometimes production is negative. Do greater uncertainties need to be

included?

P10:L29 I would expect that soils experiencing generally higher temperatures would also experience higher optimum temperatures for soil OCS uptake. Also, there's a seemingly abrupt shift in the discussion in this section, where referring back to "our finding" on L32 is a bit of a whiplash.

P11:L2-3. OCS production from autoclaved soils is assumed to be abiotic, with some sort of organic material as the substrate. In this way, OCS emissions from "dead" soils is directly related to past biological activity. Some enzymes can survive autoclaving. I am skeptical that these enzymes can then continue their OCS production for days in high temperatures and with very little water. Please do this experiment! Otherwise, this part makes it sound like only in tact enzymes can relate emissions to biological activity in dead soils. We do not need so creative a hypothesis for the argument.

Minor Comments

P2:L2 The global warming potential of OCS is roughly balanced by its "global cooling" potential, see Brühl, C., Lelieveld, J., Crutzen, P. J. and Tost, H.: The role of carbonyl sulphide as a source of stratospheric sulphate aerosol and its impact on climate, Atmos. Chem. Phys., 12(3), 1239–1253, 2012.

P3:L3 We did do a variable OCS concentration experiment in Whelan et al 2016 (the soil incubation study), without high OCS concentrations, see Fig 4 in that paper.

P4:L11 Reports of their values are scarce.

P4:L30-31 missing references. But do you really need a reference for linear regression?

P5:L22 Were there any sealants used to get the lid air-tight?

P6:L26 Is SFdry just Fdry?

P7:L22 This sentence is a lot to unpack. Please break it up.

P8:L2 The variability of the net fluxes?

P8:L22-P9:L6 This section needs a better paragraph structure. The first sentence is good. the ending is good. In between needs better vision of why each number is being reported.

P9:L16 Should be Whelan et al., 2016.

P9:L21 errant comma

P9:15 to P10:L16 The first part of this discussion has good content, but unnecessary parentheticals and some needlessly complicated sentences. Please rework.

P10:L25-26 Conrad did a follow up study that claims a second OCS soil uptake pathway at high concentrations, see citation above.

P11:L6 "Agricultural" is a better word than "arable" here. I know they're referred to as arable soils in the literature, but arable refers to soil that could support crops, where agricultural means that there are actual crops present. In all studies referred to here, there are crops present.

P12:L13-17 This overstates the case for the study. It's not clear why the relationship between N and S is now relevant where it wasn't before, or why the relationship between N inputs and S emissions constitutes a new modeling framework for atmospheric chemistry.

P12:L25 and elsewhere. Ogee 2016 model publication didn't have a production rate that wasn't redox dependent. Referring to the model via its citation might be misleading.

P13:L10 It is well known now (hopefully) that, although the Kettle 2002 study was an excellent first guess, it should not be used for global modeling studies.

P13:L12 Do you mean to have the second "modelling" there?

P13:L19-23 This sentence has a lot of information crammed into it. Please rephrase it,

perhaps breaking it up into two sentences.

P20, and elsewhere, you need a Whelan 2016a and 2016b.

P24 Figure 3 demonstrates the complexity of the analysis without adding further information. Please move this to the supplement.

P25 Figure 4, it looks like the color bar has discrete colors, but the numbers are on a continuous color spectrum? This is a little confusing, because it looks like different data might be shown on either side of the diagonal. Unless I'm misreading it, this figure only needs to present the rho's once (use either side).
* * *

---

## Author Comment (AC1) · 21 May 2018

Referee 1

Q1/ Major comments: Pg 5, line 25: You describe using dry synthetic air flows for these experiments. Will the drying of the samples using these gas flows affect the results? Or was the air humidified and it's mentioned? Or was this accounted for by adding distilled water (as described in Pg 5 line20)? It's worth clarifying this.

R1/ We agree that this important point deserves describing in more detail. The use of

dry synthetic air inevitably enhanced the drying of the soil during the experiment but to an extent that remained small enough to not influence the COS fluxes significantly. The pots were weighed before and after the gas exchange measurements and, on average, the water loss represented less than $0.2 \pm 0.005$ g of water per hour for an initial average water content of $58.7 \pm 16$ g. This represented a total reduction in volumetric soil water content of less than $0.01 \pm 0.002$ cm3 cm 3 during the full measuring sequence (16h). This water loss was small considering that we were measuring the replicates of the same soil (for a given COS concentration and temperature) every 44 minutes. In most cases, the measurements were prolonged overnight (because the sequence was automated) and this allowed us to check that the COS and CO2 fluxes of a given microcosm (at a given temperature and COS concentration) changed only marginally at >16h intervals. We have now added in the manuscript: - on page 6 line 14: "All pots were weighed before and after gas exchange measurements to calculate water loss. On average, the water loss was $0.2 \pm 0.005$ g per hour or $4 \pm 1\%$ of the initial water amount (i.e. a reduction of the volumetric water content of less than 0.01 cm3 cm 3). Additional gas exchange measurements were also performed after the end of the sequence and we could verify that the COS and CO2 fluxes of a given microcosm (at a given temperature and COS concentration) were not significantly different between the first and second sequences (16h apart; see below). This was a clear indication that the small water loss during the duration of the gas exchange measurements did not impact significantly the COS and CO2 fluxes." - on page 7 line 3: we have added that the triplicate measurements were performed "every 44 minutes to partially take into account the possible variability caused by the small water loss".

Q2/ I like the layout of the Discussion with the driving question at the start but it might have even more impact with the answers to these questions instead. Might be nice to follow that through the Discussion rather than the current headers.

R2/ We have now changed the headers of the discussion to make them more informative: - header 4.1 is now "COS production rates measured on dry soils are a reasonable

proxy for those occurring in moist soils" instead of "Are COS production rates measured on dry soils a reasonable proxy for those occurring in moist soils?" - header 4.2 is now "Soils generally act as COS sinks at cool temperatures but become COS sources rapidly upon warming" instead of "Relative importance of gross COS production to the net soil COS flux" - header 4.3 is now "Soil COS production rates also increase with soil N content and mean annual precipitation" instead of "Drivers and mechanisms of COS production across European soils from different biomes and land use" - header 4.4 is now "The soil COS uptake rate constant increases with soil microbial content and has a small temperature sensitivity" instead of "Drivers and mechanisms of COS uptake by soils"

Q3/ I feel like the abstract could do with a nice tie-up/bigger implications type sentence. Could you include something from the conclusions (maybe the N impact?)?

R3/ We have rephrased the last sentence of the abstract: "Collectively our findings suggest a strong interaction between soil nitrogen and water cycling on COS production and uptake, providing new insights on how to upscale the contribution of soils to the global atmospheric COS budget."

Q4/ Pg 1, line 25: Not essential, but could you edit the last sentence to be more specific? It seems really vague.

R4/ As suggested above, this last sentence of the abstract has been edited to reinforce the idea that bringing in the N cycle is key to upscale the contribution of soils to the global COS atmospheric budget.

Q5/ Pg 4, line 30: references missing.

R5/ Corrected.

Q6/ Pg 5, line 8: Were the soil samples chilled for shipping? Or do you have any idea of the temperature history of the samples?

R6/ The soil samples were not chilled for shipping as they were sent by regular mail, so

we do not know exactly the temperature history of the sample before reception. This is why they were all stored at 4°C immediately after reception and re-acclimated for 2 weeks at 18°C before the gas exchange measurements. We have now added in the manuscript: - on page 5 line 8: "The first ten centimeters of soil were collected at three locations at each site and sealed in plastic bags and sent to INRA Bordeaux after collection with no special requirements imposed for the transportation of the soil samples. Upon reception, the different soils were sieved using a 4mm mesh, homogenised and stored at 4°C."

Q7/ Pg 8, line 2: Do you mean the bias in the blank? The uncertainty on the blank value is 0.24 pmol.

R7/ Yes this is what we meant. We have changed this sentence page 8, line 26 to: "In comparison, the blank was not significantly different from zero with mean COS flux values of $0.11 \pm 0.24$ pmol m 2 s 1."

Q8/ Figure 1: I know it looks nice to have the fluxes in order but I'm trying to visualize what this is actually telling us. Would it be worthwhile grouping them by biosphere so they can be applied to other studies a little easier? So Boreal forest vs Peatland. Mediterranean Orchards vs grass vs forest and Temperate grass vs forest? Or something like that. And is grassland FR really boreal? Or do you mean alpine? Not in Table 1 so hard to tell. You could include the full site labels too. Could you also add some gridlines so it's easier to see what the labels are matching to?

R8/ We agree this way of presenting could be more informative. We have thus re-drawn the figure (Fig.1) and ordered the fluxes by biomes and land uses, and added the full site labels.

Q9/ Pg 11 line 4: I think this section could be tightened a little.

R9/ We have rewritten this section, page 12, line 2: "These higher compensation points might be explained by the warmer temperatures expected in this type of climate that

should favour COS production over consumption (Fig. 6), shifting the compensation point to higher COS concentration values and even causing some soils to become net COS emitters. Because the temperature sensitivity of the production rate is always larger than that of the hydrolysis constant, a potential shift in the optimum temperature of COS uptake would not be enough to offset the larger production rates at the higher temperatures. This relatively greater temperature sensitivity of COS production rates found in our experiment are also consistent with a number of previous studies reporting the temperature sensitivities (Q10) of production-dominated net COS soil fluxes in the range of 1.7 to 3.3 (Maseyk et al., 2014; Saito et al., 2002; Saito et al., 2002; Whelan and Rhew, 2015, 2016). Altogether, our results show that soil COS production (and its contribution to the net COS flux) varies across different biome and temperature regimes and must be accounted for when performing atmospheric COS budgets."

Q10/ Figure 3: I must admit that I don't follow Fig 3. What do the dimensions (x vs y axis) represent? Is it really necessary?

R10/ Each axis represents a linear combination of the overall variables used in the PCA (the so-called principal components). This graph indicates that the soils are mainly differentiated by soil C and N content and microbial biomass C and N (these 4 variables co-varying), and that the COS source is mainly related to them. However, this last (and principal) information is also shown in Figure 4. Therefore, we have moved Figure 3 to the supplementary data (now Figure S5).

Q11/ Could you include Fig S4 in the main text? It's just missing the A and N labels for fertilizers. Is the CH Grassland not an agricultural land (Fig S5)? I's surprised you can find an unfertilized grassland in Europe!

R11/ As suggested, we have now moved Fig S4 to the main text (in place of the old Figure 3). We also grouped the results by biomes and indicated the associated land use. The notation "Fertilised" in this dataset is used to differentiate, for a given site, experimental parcels where N fertilisers are being experimentally manipulated compared

with those at the same site that did not receive additional N fertilisers. The CH Grassland with a high COS source that the referee probably refers to is Fruebuel, a site that did not have different experimental parcels with different N treatments thus, although it is likely fertilised by animals that graze the site, it is not specified in the site name as a site with a specific experimental manipulation. In order to avoid such confusion we now give on the x-axis of Figure 3 the full site name, and add this information in the caption of table 1.

Q12/ Fig 4: Is there a reason for the order in Fig 4? Would it make more sense to keep the soil type (clay, silt, sand) together and microbial properties (Soil N, Soil C, Soil P, MBC and MBN) together? Then derived properties like PCOS (production), LCOS (loss) and k18. Some of the labels are not adequately explained. What do you mean by Redox, Q10k, etc. I know they are explained in the text but make sure the figures can be read independently.

R12/ We agree that the order of the variables were not grouped and we have now changed it to group similar variables (i.e. texture, biochemical variables, COS production, COS consumption...) together. We also used the exact same symbols as in the main text and re-stated their meaning in the figure caption.

Q13/ Fig 5 and 6 could go in the supplement.

R13/ We agree that Fig. 5 is not essential but Fig. 6 is clearly an important result that should stay in the main text. Because the number of figures is still small (6) we did not feel the need to move more figures to the supplementary material.

Q14/ Table 1: Could you add the altitude of the sites? And maybe the annual mean soil temperature and moisture at each site if you have that data? Can you also include an explicit Fertilizer or not column?

R14/ For each site, we have now added the altitude, mean annual temperature and precipitation (MAT and MAP) values, according to literature values or websites describing

each experimental plot. The two names in bold (Rosinedal_Fert and Laqueuille_Fert) are the only experimental parcels where N fertilizers have been added and are compared to adjacent parcels without this N fertilization addition. This is now explained in the caption of the Table.

Referee 2. M.E. Whelan.

This is a well-conceived set of experiments to further our understanding of oxic soil OCS exchange. The approach to calculate the hydrolysis constant is commendable. It's also nice to see the 1996 Lehmann and Conrad study getting more use. Please note the follow up study in 2000 by Conrad and Meuser, "Soils contain more than one activity consuming carbonyl sulfide" Atmospheric Environment, 21, 3635-3639.

R/ Thank you for this very positive comment. The article of Conrad and Meuser, 2000 is now cited in section 4.2.

Major Comments

Q1/ P3:L22-25 I'm not sure how theta going to zero results in the simplified equation presented here. Are there some assumptions that need to be spelled out?

R1/ There is no extra assumption. It is a consequence of the tanh function. We have added this information to make the statement less difficult to follow: "Noting that tanh(ax)/x = a if x→0, when soil moisture tends to zero (θ→0), Eq. 1 simplifies to Fdry = −bPdryzmax, where Fdry and Pdry represent the net COS flux F and the COS production rate P of a air-dry soil, respectively."

Q2/ P5:L5-20 Please add more detail. For either method, were the jars partially sealed, generating higher CO2 levels than ambient?

R2/ During the 2-week incubation period in the climate chamber, the jars were not sealed and the concentration of COS and CO2 were semi-controlled by circulating air through the climate chamber with CO2 and COS concentrations around 400 ppm and 500 ppt, respectively. Thus the CO2 concentration remained very close to ambient

levels and to the concentration used for the gas exchange measurements (for COS, we varied the concentration from 100 to 1000 ppt during the gas exchange measurements, as described in the Methods). Therefore, at no time were the microcosms exposed to $CO_2$ levels much higher than in the ambient air. We have now added this information in the revised manuscript: Page 5, line 21: "During the 2-week incubation period in the climate chamber, the microcosms were not sealed and the air circulating in the climate chamber had $CO_2$ and COS concentrations controlled around 400 ppm and 500 ppt, respectively, i.e., close to ambient levels. The same $CO_2$ concentration was used on the air inlet during the gas exchange measurements, so that the microcosms were never exposed to $CO_2$ levels much higher than in ambient air."

Q3/ Were any sensitivity studies performed – for example, did you find that incubating the soils for less than 2-3 days or two weeks led to different results?

R3/ Unfortunately we did not perform such a sensitivity study only one on the Birch effect described in our response below.

Q4/ For method 2, were these soils kept in the dark as well?

R4/ Yes they were. We have added this information in the manuscript page 5 line 25.

Q5/ When soils were air dried, were they put into a jar or spread out in a pan for a more even drying?

R5/ To dry them, the soils were spread in aluminum trays and mixed regularly (every 2-3 days) to reach an even drying. We have added this information in the manuscript page 5 line 16: "One batch was air-dried by spreading soil in a tray and regularly mixing every 2-3 days for 1-2 weeks before being measured to estimate the air-dried COS production rate (Pdry) hereafter referred to as "dry"."

Q6/ Sieving is an important choice here, too. Litter plays a role in surface OCS fluxes, sometimes contributing nearly all of the OCS uptake. Sieving removes most of the litter and soil structure. While we can't have everything in our experiment vary, it would be

worth justifying the method approach a bit more.

R6/ We agree that sieving the soils and removing the litter influences the soil microbial community and the soil C and N dynamics compared to those of non-sieved soils (Thomson et al., 2010; Effects of sieving, drying and rewetting upon soil bacterial community structure and respiration rates, Journal of Microbiological Methods 83: 69-73). We also acknowledge that these changes in soil structure and organic content will also modify the COS fluxes compared to those that would be observed on intact soils or in the field. However our aim was to understand the drivers of COS consumption and production. For this we needed very homogeneous soil samples, both in terms of structural components and environmental variables (moisture, N contents...). The main hypothesis is that the drivers identified in our study would be the same as in undisturbed soils. We also followed the advice of Thomson et al. (2010) that recommended to sieve soils when fresh to minimize the impact on microbial activity. We have now added this information in the manuscript.

Page 5, line 11: "The sieving was performed to ensure a representative sample of only soil, to avoid introducing any additional and (uncontrolled) plant litter effects that could potentially introduce variability between sample replicates and complicate the interpretation of the net and gross COS fluxes. We justify this experimental choice as our overall goal was to derive and validate a model of soil COS fluxes regulated by commonly quantified soil physical, chemical and biological characteristics. Our main hypothesis is that the drivers identified in our study would still be applicable in undisturbed soils. Sieved soils were then separated into two batches. Âż

Q7/ Regarding maintaining soil moisture by adding water – if the soil has dried out enough (probably crossing some threshold that has yet to be described) and water is added, the soil can experience a dramatic increase in OCS uptake that takes several hours to days to recover from. This is akin to the "Birch effect" for OCS. Please mention how much water was typically added to the soils. Did you see any decay curve in the soils that were maintained this way?

R7/ Everything was done to minimize measurement during the so-called "Birch effect" especially during the gas exchange measurements as this could have varied dramatically between soil types depending on their initial moisture content when arriving in the lab. The largest water addition was performed before the incubation period (especially on soils that were shipped from very dry places). Tests where COS and CO2 gas exchange were measured regularly after this initial water addition showed that, after 2 weeks of incubation, we were well outside the decay curve of the Birch effect. Throughout the incubation period, extra water additions were also performed (about 2-5 g of distilled water for an average water content of 58.7 $\pm$ 16 g), and the last water addition was performed 24h before the gas exchange measurements to minimise the Birch effect. This is now explained in the manuscript.

Page 5, line 32 : "Care was taken to avoid measuring the so-called "Birch effect" (Jarvis et al., 2007) during gas exchange measurements. Because the largest water addition was performed just before the incubation period (especially on soils that were shipped from very dry places) all gas exchange measurements were delayed for 2 weeks to ensure fluxes had stabilised during the incubation period and that they were outside the decay curve of the Birch effect. Throughout these 2 weeks, microcosms were kept unsealed in the dark and the moisture contents were monitored gravimetrically every two days whereupon, extra but small, water additions were made (about 2-5 g of distilled water for an average water content of 58.7 $\pm$ 16 g) and no later than 24h before the start of the gas exchange measurements."

Q8/ Also, 18 hours is a long time to have dry air run over soils without substantial water loss. Were soils checked and re-watered during the incubations?

R8/ We weighed the pots before and after the gas measurement to calculate the water loss. On average, they lost 0.2 $\pm$ 0.005g of water per hour representing a final loss of 4 $\pm$ 1% of the initial water amount (i.e. a reduction of the volumetric water content of less than 0.01 cm3 cm 3). Additional gas measurements were also performed after the end of the sequence, and the values of COS and CO2 of the first and second

measurements (16h apart; see below) were not significantly different, demonstrating that the small water loss during the gas exchange measurements did not impact significantly the measurements. This information is now added to the manuscript in section 2.3 (see response to first comment of reviewer 1).

Q9/ P7:L10 Is 5 degrees sufficient to calculate a meaningful Q10? Also, OCS uptake rates tend to exhibit a temperature optimum. The Q10 idea links the rate of reaction with a constant increase in rate with increase in temperature. Please justify the use of Q10. It would be good to know the natural variation in temperature of the sites as well.

R9/ We agree that a 5°C difference is small to characterise precisely a temperature response. However, it was a compromise to minimise the time spent on a sequence of measurements as we had to make sure that the new set temperature was reached and fully homogenised throughout all microcosms. The temperature range explored (18-23°C) was also well outside any enzymatic temperature optimum (expected to be at >25°C), thus justifying the use of a Q10 response for both COS production and consumption processes. Actually, the temperature response of plant CA is often described by such a Q10 response (e.g. Burnell, J. N. and Hatch, M. D.: Low bundle sheath carbonic anhydrase is apparently essential for effective C4 pathway operation, Plant Physiology, 86(4), 1252–1256, 1988). Also, because our study was gathering soils from different biomes, with quite large differences in mean annual temperatures (added to Table 1), it seemed necessary to explore a range of temperatures common to all of these biomes.

Q10/ P7:L15 I was not aware that soil redox potential could still yield a valid measurement after 2 weeks. How do you think this variable changed during the incubations themselves?

R10/ We agree that soil redox potential measurements are highly variable and rapidly perturbed by external factors. We thus measured the soil redox potential just after the gas exchange measurements in order to evaluate whether this highly variable property

could be related to the COS fluxes, as was suggested in previous studies (Devai, I. and Delaune, R. D.: Formation of volatile sulfur compounds in salt marsh sediment as influenced by soil redox condition, Organic Geochemistry, 23(4), 283–287, 1995.). However we are well aware that redox potential is not a fixed characteristic of the soils. We also measured the redox potential on dry and wet soils, and in both cases, the measurements were well replicated and, although the values were different between dry and wet soils, the differences between the soils were well conserved and the two measurements were linearly correlated (slope of 1.16 and R2 of 0.6). This gave us confidence to use redox potential as a soil property to explain our gas exchange data. However, unexpectedly, the correlation with the COS fluxes was weak, even when only COS production was considered, so we did not discuss the results further. However, we still give the redox potential values because redox potential could be a useful integrative tool (Husson, 2012, Redox potential and pH as drivers of soil/plant/microorganism systems: a transdisciplinary overview pointing to integrative opportunities for agronomy. Plant Soil 362:389-417).

Q11/ P8:L10 What is going on with the green points in Fig 2 that have a wide spread? Also, it appears that sometimes production is negative. Do greater uncertainties need to be included?

R11/ We agree that, at intermediate soil N contents (green points in Fig. 2a) the agreement between Pmoist and Pdry is the weakest, but we do not have a clear explanation for this. Upon investigation we found that three of the soils had much higher Pmoist than Pdry . We believe this may indicate that the air-dried soils may not have been completely dry. This hypothesis is partially supported by the persistence of a small net CO2 flux, indicating that some uptake of COS could still be possible thus leading to the negative net COS fluxes. Despite these complications, overall the linear regressions stood. We have added page 9, line 6 " Upon investigation we found that some soils having much higher Pmoist than Pdry may not have air-dried soils completely dry. This hypothesis is partially supported by the persistence of a small net CO2 flux (Table S1)

[Figure]

for these soils, indicating that some uptake of COS could still be possible contributing to some of the dispersion in the data."

Q12/ P10:L29 I would expect that soils experiencing generally higher temperatures would also experience higher optimum temperatures for soil OCS uptake. Also, there's a seemingly abrupt shift in the discussion in this section, where referring back to "our finding" on L32 is a bit of a whiplash.

R12/ Even with higher optimum temperatures for COS uptake, the temperature sensitivity of the production rate would still be much higher than the uptake, thus shifting the compensation points to higher COS concentration values.

Q13/ P11:L2-3. OCS production from autoclaved soils is assumed to be abiotic, with some sort of organic material as the substrate. In this way, OCS emissions from "dead" soils is directly related to past biological activity. Some enzymes can survive autoclaving. I am skeptical that these enzymes can then continue their OCS production for days in high temperatures and with very little water. Please do this experiment! Otherwise, this part makes it sound like only intact enzymes can relate emissions to biological activity in dead soils. We do not need so creative a hypothesis for the argument.

R13/ We agree and now rephrased this section of the discussion to clarify that, even if totally abiotic, COS production from soils should be related to past biological activities, without the need to evoke any enzymatic activity. In specially, we have added Page 13, line 10: "Although our results cannot rule out any of the above mechanistic hypotheses, our results and previous studies indicate overall that the COS emission rates of air-dried and autoclaved soils are related to past biotic activity and in particular the soil nitrogen status. Âż

Minor Comments

Q14/ P2:L2 The global warming potential of OCS is roughly balanced by its "global cooling" potential, see Brühl, C., Lelieveld, J., Crutzen, P. J. and Tost, H.: The role

of carbonyl sulphide as a source of stratospheric sulphate aerosol and its impact on climate, At- mos. Chem. Phys., 12(3), 1239–1253, 2012.

R14/ We have added this reference and clarified the text regarding the GWP of COS. Page 2, line 2 : " Carbonyl sulphide (COS) is a powerful greenhouse gas whose atmospheric concentration has varied considerably during the Earth's history (Ueno et al., 2009). In the present day stratosphere, COS photolysis contributes to the formation of aerosol particles that also cool the planet, consequently offsetting the global warming potential of COS (Brühl et al., 2012)."

Q15/ P3:L3 We did do a variable OCS concentration experiment in Whelan et al 2016 (the soil incubation study), without high OCS concentrations, see Fig 4 in that paper.

R15/ We have changed the text and added this reference here in the text. Page 3, line 3 : Âń Because this alternative approach requires the measurement of net COS fluxes at different atmospheric COS concentrations, it cannot be easily implemented in the field without large artefacts (Castro and Galloway, 1991; Mello and Hines, 1994), but it is well adapted to measurements on soil microcosms (Lehmann and Conrad, 1996; Conrad & Meuser, 2000; Whelan et al., 2016b). So far very few studies have implemented this approach, thus the partitioning of COS fluxes at ambient concentrations still remains poorly explored (Whelan et al., 2017). Âż

Q16/ P4:L11 Reports of their values are scarce.

R16/ We have changed the sentence in the manuscript.

Q17/ P4:L30-31 missing references. But do you really need a reference for linear regression?

R17/ We have now added the information page 5, line1, notably concerning the "fzero" function in the Pracma R package (Brochers, 2017), and the reference was also added in reference list.

Q18/ P5:L22 Were there any sealants used to get the lid air-tight?

R18/ No sealants were used inside the chamber. We have customised glass jars and lids that have been very finely sanded to provide a glass on glass seal. The glass seal is maintained by screwing the lid of the jar over the glass lid. The airstreams are facilitated by two stainless steel Swagelok fittings sealed with PTFE washers. The system was shown to be COS neutral (and is systematically tested by including a blank jar in each measuring sequence on a randomly selected set of inlet and outlet lines). The temperature probe was fixed in the same way via a Swagelok connector. We have added this information in the manuscript page 6 line 7 :" Glass soil microcosms were equipped with customised screw-tight glass lids cut to the exact size of the microcosm area and finely ground to provide a glass-on-glass seal that was held in place by screwing the threaded metal lid onto the jar over the glass panel. The lids were equipped with two stainless steel Swagelok$^{®}$ (Swagelok, Solon, OH, USA) fittings to connect to the 1/8" Teflon inlet and outlet lines of the measurement system. Sealing was ensured using PTFE washers that were previously tested and shown not to emit COS. Âż

Q19/ P6:L26 Is SFdry just Fdry?

R19/No, SFdry stands for the product of the soil surface area S (defined above in Eq. 4) and Fdry.

Q20/ P7:L22 This sentence is a lot to unpack. Please break it up.

R20/ The sentence is now divided into three different sentences.

Q21/ P8:L2 The variability of the net fluxes?

R21/ Yes it is the variability of the net fluxes. We have added page 8, line 26 : "All moist soils were net COS sinks at 18°C, with net fluxes ranging in magnitude from 7.66 to 0.78 pmol m 2 s 1 (Fig. 1)."

Q22/ P8:L22-P9:L6 This section needs a better paragraph structure. The first sentence is good. the ending is good. In between needs better vision of why each number is being reported.

[Figure]

R22/ The section was restructured.

Q23/ P9:L16 Should be Whelan et al., 2016.

R23/ Corrected.

Q24/ P9:L21 errant comma

R24/ Removed.

Q25/ P9:15 to P10:L16 The first part of this discussion has good content, but unnecessary parentheticals and some needlessly complicated sentences. Please rework.

R25/ Done

Q26/ P10:L25-26 Conrad did a follow up study that claims a second OCS soil uptake pathway at high concentrations, see citation above.

R26/ This article is now cited in the text page 11, line 28 : "This apparent contradiction might be explained by the much higher and wider range of COS concentrations (60-410 ppb) explored by Lehmann & Conrad (1996) where different COS consumption processes might take place (e.g. physio-sorption; Conrad and Meuser, 2000)."

Q27/ P11:L6 "Agricultural" is a better word than "arable" here. I know they're referred to as arable soils in the literature, but arable refers to soil that could support crops, where agricultural means that there are actual crops present. In all studies referred to here, there are crops present.

R27/ We agree and replaced "arable" with "agricultural" throughout the manuscript.

Q28/ P12:L13-17 This overstates the case for the study. It's not clear why the relationship between N and S is now relevant where it wasn't before, or why the relationship between N inputs and S emissions constitutes a new modeling framework for atmospheric chemistry.

R28/ We rephrased the last few sentences of this paragraph, page 13, line 23: "Thus

ecosystems exposed to higher nitrogen inputs either naturally or by enhanced fertilisation may be creating plant and soil organic matter that contains relatively more N- and S-containing precursors such as amino acids and proteins that then become available substrates in soils for temperature-sensitive abiotic degradation. Further studies investigating the link between soil N inputs and soil COS fluxes would now be useful to assess whether total soil N and soil microbial N biomass as traits could be helpful integrated predictors of how soil COS production and uptake rates, respectively vary across large spatial scales irrespective of whether the underlying mechanism of COS production is abiotic or biotic in nature. Meanwhile parallel studies clarifying the mechanistic processes underlying the production will aid the development of models attempting to describe dynamically the instantaneous exchange between soils and the atmosphere and their link to climate, vegetation type and management regime. Âż

Q29/ P12:L25 and elsewhere. Ogee 2016 model publication didn't have a production rate that wasn't redox dependent. Referring to the model via its citation might be misleading.

R29/ Given the range of redox potential explored here, the formulation proposed by Ogée et al. would be insensitive to redox potential changes, so the model could still hold. The main result here is that the base production rate P25 is related to total N content, a result not incorporated in Ogée et al. 2016

Q30/ P13:L10 It is well known now (hopefully) that, although the Kettle 2002 study was an excellent first guess, it should not be used for global modeling studies.

R30/ We agree but because the values provided by Kettle et al (2002) are within the range of values proposed by our studies, we did not feel it necessary to remove it from the list.

Q31/ P13:L12 Do you mean to have the second "modelling" there?

R31/ Corrected.

Q32/ P13:L19-23 This sentence has a lot of information crammed into it. Please rephrase it, perhaps breaking it up into two sentences.

R32/ This sentence has been split and simplified. Page 15, line2 : "In particular we showed that COS hydrolysis rates were linked to microbial C biomass whilst COS production rates were linked to soil N content and MAP. In addition both of these gross COS fluxes exhibited distinctly different temperature and moisture sensitivies. These different soil proprieties should now be explored more deeply to determine their added-value in the prediction of soil COS fluxes and their ability to reconcile the contribution of soil COS fluxes to the atmospheric COS budget. Âż

Q33/ P20, and elsewhere, you need a Whelan 2016a and 2016b.

R33/ Corrected.

Q34/ P24 Figure 3 demonstrates the complexity of the analysis without adding further information. Please move this to the supplement.

R34/ We have moved this figure to the supplementary information.

Q35/ P25 Figure 4, it looks like the color bar has discrete colors, but the numbers are on a continuous color spectrum? This is a little confusing, because it looks like different data might be shown on either side of the diagonal. Unless I'm misreading it, this figure only needs to present the rho's once (use either side).

R35/ The figure has been entirely re-drawn, also to satisfy the other reviewer's comments.

Please also note the supplement to this comment:
https://www.atmos-chem-phys-discuss.net/acp-2017-1229/acp-2017-1229-AC1-supplement.pdf

———————————————

2018.